# Morpho-Phylogenetic Evidence Reveals Novel Pleosporalean Taxa from Sichuan Province, China

**DOI:** 10.3390/jof8070720

**Published:** 2022-07-09

**Authors:** Xian-Dong Yu, Sheng-Nan Zhang, Jian-Kui Liu

**Affiliations:** Center for Informational Biology, School of Life Science and Technology, University of Electronic Science and Technology of China, Chengdu 611731, China; yuxiandong@std.uestc.edu.cn (X.-D.Y.); zhangshengnan@uestc.edu.cn (S.-N.Z.)

**Keywords:** 6 new taxa, Dothideomycetes, multi-gene, phylogeny, taxonomy

## Abstract

Pleosporales is the largest and most morphologically diverse order in Dothideomycetes, including a large proportion of saprobic fungi. During the investigation of microfungi from decaying wood in Sichuan Province, several novel fungal taxa of asexual and sexual morphs were collected, identified, and well-described. Phylogenetic analyses based on SSU, ITS, LSU, *RPB2* and *TEF1**α* gene sequences suggested that these new taxa were related to Pleosporales and distributed in five families, *viz.* Amorosiaceae, Bambusicolaceae, Lophiostomataceae, Occultibambusaceae and Tetraplosphaeriaceae. The morphological comparison and molecular phylogeny evidence justify the establishment of six new taxa, namely *Bambusicola guttulata* sp. nov., *Flabellascoma sichuanense* sp. nov., *Neoangustimassarina sichuanensis* gen. et sp. nov., *Occultibambusa sichuanensis* sp. nov. and *Pseudotetraploa bambusicola* sp. nov. Among them, *Neoangustimassarina* was introduced as the second sexual morph genus in Amorosiaceae; *Bambusicola guttulata*, *O. sichuanensis* and *P. bambusicola* were isolated from bamboos, which contributed to the diversity of bambusicolous fungi. The detailed, illustrated descriptions and notes for each new taxon are provided, as well as a brief note for each family. The potential richness of fungal diversity in Sichuan Province is also discussed.

## 1. Introduction

Pleosporales is the largest order in the class Dothideomycetes [1], and its members are found worldwide on a variety of host plants as epiphytes, endophytes, saprobes and parasites [2,3,4]. In addition, they are commonly found in terrestrial, marine and freshwater habitats [5,6,7]. Some of them produce secondary metabolites that can serve as a basis for developing new antimicrobials, agrochemical pesticides and other useful compounds [8].

The Pleosporales was invalidly introduced by Luttrell [9], and later revised by Barr [10], based on the family Pleosporaceae with the type species *Pleospora herbarum* [11]. Members of Pleosporales usually have perithecioid ascomata, cellular pseudoparaphyses, bitunicate and fissitunicate asci, and various shaped, aseptate or septate ascospores, with or without a gelatinous sheath [12,13,14], and their asexual morphs are coelomycetes and hyphomycetes [12,13]. For example, asexual morphs of *Bambusicola* and *Pseudotetraploa* are the most common forms of coelomycetes and hyphomycetes in Pleosporales, respectively. Zhang et al. listed 26 families in Pleosporales [12], while Hyde et al. revised Pleosporales and accepted 41 families [13]. Hongsanan et al. redefined the families of Dothideomycetes and accepted 91 families in Pleosporales based on morphology and multigene analysis [15]. Currently, Pleosporales consists of approximately 91 families and 655 genera (including 41 genera *incertae sedis*) [16].

Fungi have a broad geographical distribution and diversity comparable to plants and other organisms [17,18]. However, the fungal kingdom, in general, is less well-documented than the plant kingdom in terms of the number of species [19]. As one of the biodiversity hotspots in China, Sichuan Province (located in southwestern China) has a variety of complex topography (mountains, hills, plains, basins and plateaus) and climate conditions, and these are important factors contributing to the biodiversity [20]. However, little research on fungi has been carried out in this area; meanwhile, the highly variable climate and lush vegetation are shown to have an important influence on fungal diversity. Therefore, Sichuan Province is believed to have a large amount of hidden fungal diversity to be explored and discovered [21,22].

During a survey of micro-fungi from decomposing wood in Sichuan Province, China, a series of interesting asexual and sexual fungi were collected. In this study, we aim to describe these new findings and contribute fungal diversity to China. The multi-gene phylogeny integrated with morphological comparison was carried out to determine the classification of these new collections. One new genus and five new species are introduced, and the establishment of these new taxa is justified by morphology and phylogenetic evidences.

## 2. Materials and Methods

### 2.1. Isolation and Morphological Examination

Fungi associated with decaying wood were collected from Sichuan Province, China in 2021. Specimens were placed in envelopes and taken to the laboratory. Fungal colonies and fruiting bodies were observed using Motic SMZ 168-B. Fungal structures were examined and photographed by using a Nikon ECLIPSE Ni-U compound microscope fitted with a Nikon DS-Ri2 digital camera. The detailed morphological examination approaches used in this paper were generally based on Senanayake et al. [23]. Single spore isolations were made following the method in Senanayake et al. [23]. Measurements were made with the Tarosoft (R) Image Framework program v. 0.9.7, following the procedures outlined by Liu et al. [24]. Photo plates representing fungal structures were processed in Adobe Photoshop CS6 software (Adobe Systems Inc., San Jose, CA, USA). Herbarium specimens (dry branches with fungal material) were deposited in the herbarium of Cryptogams, Kunming Institute of Botany Academia Sinica (HKAS), Kunming, China and the herbarium of the University of Electronic Science and Technology (HUEST), Chengdu, China. The isolates obtained in this study were deposited in China General Microbiological Culture Collection Center (CGMCC), Beijing, China and the University of Electronic Science and Technology Culture Collection (UESTCC), Chengdu, China. The names of the new taxa were registered in MycoBank [25].

### 2.2. DNA Extraction, PCR Amplification and Sequencing

A Trelief TM Plant Genomic DNA Kit (Beijing TsingKe Biotech Co., Ltd., Beijing, China) was used to extract total genomic DNA from fresh mycelia, according to the manufacturer’s instructions. DNA amplification was performed by polymerase chain reaction (PCR). SSU, ITS, LSU, *RPB2* and *TEF1α* gene regions were amplified using the primer pairs NS1/NS4, ITS5/ITS4, LR0R/LR5, fRPB2-5F/fRPB2-7cR and 983F/2218R, respectively, [26,27,28,29]. The amplification reactions were performed in 25 μL PCR mixtures containing 22 μL PCR MasterMix (Green) (TsingKe Co., Beijing, China), 1 μL DNA template and 1 μL of each primer (10 µM/L). The PCR thermal cycle program for SSU, ITS, LSU, *RPB2* and *TEF1α* amplification were listed in Table 1. PCR products were checked on 1% agarose electrophoresis gels stained with Gel Red. The sequencing reactions were carried out with primers, mentioned above, by Beijing Tsingke Biotechnology Co., Ltd., Chengdu, China.

### 2.3. Phylogenetic Analyses

The chromatograms of the new sequences obtained in this study were viewed in Finch TV Version 1.4.0 (https://digitalworldbiology.com/FinchTV (accessed on 22 September 2021)). The BLAST searches were performed for finding similar sequences that match our data. A concatenated dataset of the SSU, ITS, LSU, *RPB2* and *TEF1α* sequences were used for phylogenetic analyses with the inclusion of reference taxa from GenBank (Table 2). The sequences were aligned by using the online multiple-alignment program MAFFT v.7 (http://mafft.cbrc.jp/alignment/server/ (accessed on 5 January 2022)) [30], and the alignment was manually optimized in BioEdit v.7.0.9 [31]. Each gene dataset was concatenated by Mesquite v. 3.11 (http://www.mesquiteproject.org/ (accessed on 15 April 2022)) for multi-gene phylogenetic analyses. Maximum likelihood (ML) and bayesian inference (BI) were carried out as detailed in Dissanayake et al. [32]. The programs used in this study are RAxMLGUI v. 1.0 [33], PAUP v.4.0b10 [34], Mr Modeltest 2.3 [35] and MrBayes v. 3.1.2 [36,37]. The phylogenetic tree was visualized by FigTree v.1.4.0 (http://tree.bio.ed.ac.uk/software/figtree/ (accessed on 15 April 2022)).

## 3. Results

### 3.1. Phylogenetic Analyses

Five gene loci SSU, ITS, LSU, *RPB2* and *TEF1α* were used to determine the phylogenetic placement of the new collections. The concatenated matrix comprised 124 taxa with a total of 4633 characters (SSU: 1021 bp; ITS: 684 bp; LSU: 904 bp; *RPB2*: 1031 bp; *TEF1α*: 993 bp) including gaps. Maximum likelihood (ML) and Bayesian inference (BI) analyses were carried out to infer phylogenetic relationships. The best scoring ML tree (Figure 1) was selected to represent the relationships among taxa, in which a final likelihood value of –51844.747390 is presented. The matrix had 2418 distinct alignment patterns. Estimated base frequencies were as follows: A = 0.245989, C = 0.250069, G = 0.270993, T = 0.232949; substitution rates AC = 1.588007, AG = 3.646881, AT = 1.331328, CG = 1.142338, CT = 7.560715, GT = 1.000000. GTR + I + G is the best-fit model selected by AIC in MrModeltest based on each gene (SSU, ITS, LSU, *RPB2* and *TEF1α*), which was used for maximum likelihood and Bayesian analysis. Six simultaneous Markov chains were run for 1,970,000 generations and trees were sampled every 1000 generations and 1970 trees were obtained. The first 394 trees representing the burn-in phase of the analyses were discarded, while the remaining 1576 trees were used for calculating posterior probabilities in the majority rule consensus tree (critical value for the topological convergence diagnostic is 0.01).

The newly obtained isolates were grouped with pleosporalean families of Amorosiaceae, Bambusicolaceae, Lophiostomataceae, Occultibambusaceae and Tetraplosphaeriaceae. Two isolates of *Neoangustimassarina sichuanensis* (CGMCC 3.20937 and UESTCC 22.0001) formed a distinct, well-supported clade (84% ML/1.00 BYPP) in Amorosiaceae. Two isolates of *Bambusicola guttulata* (CGMCC 3.20935 and UESTCC 22.0002) were nested in the genus *Bambusicola* in Bambusicolaceae. Two isolates of *Flabellascoma sichuanense* (CGMCC 3.20936 and UESTCC 22.0003) clustered with *Flabellascoma* in Lophiostomataceae. Two strains of *Occultibambusa sichuanensis* (CGMCC 3.20938 and UESTCC 22.0004) formed a distinct branch within Occultibambusaceae, which was closely related to *Occultibambusa hongheensis* (KUMCC 21-0020), *O. maolanensis* (GZCC 16-0116), and *Versicolorisporium triseptatum* (JCM 14775 and NMX1222) with statistical support (100% ML/1.00 BYPP). The other two strains of *Pseudotetraploa bambusicola* (CGMCC 3.20939 and UESTCC 22.0005) grouped with *Pseudotetraploa* species in Tetraplosphaeriaceae.

### 3.2. Taxonomy

**Pleosporales** Luttr. ex M.E. Barr, *Prodromus to class Loculoascomycetes*: 67 (1987)

**Amorosiaceae** Thambug and K.D. Hyde, *Fungal Diversity* 74: 252 (2015)

*Notes*: Amorosiaceae was established by Thambugala et al. [38] and typified by *Amorosia* Mantle and D. Hawksw., which is characterized by micronematous to semi-macronematous conidiophores, integrated, terminal, or intercalary, monoblastic conidiogenous cells, elongate-clavate and 3–4-septate conidia [39]. Six genera were accepted in the family, *viz. Alfoldia* D.G. Knapp, Imrefi and Kovács, *Amorosia* Mantle and D. Hawksw., *Amorocoelophoma* Jayasiri, E.B.G. Jones and K.D. Hyde, *Angustimassarina* Thambugala, Kaz. Tanaka and K.D. Hyde, *Neothyrostroma* Crous and *Podocarpomyces* Crous [38,39,40,41,42]. *Angustimassarina* is the only genus in the family that represents the sexual morph [38]. Herein, we introduce the second genus with a sexual morph to Amorosiaceae.

***Neoangustimassarina*** X.D. Yu and Jian K. Liu, gen. nov.

*Type species: Neoangustimassarina sichuanensis* X.D. Yu, S.N. Zhang and Jian K. Liu

*MycoBank*: MB 843716

*Etymology*: The name refers to the similarity to the genus *Angustimassarina*.

*Saprobic* on dead wood in terrestrial habitat. **Sexual morph**: *Ascomata* solitary, scattered, immersed, visible as pale brown, circular cap with a small central black dot, subglobose, uniloculate. *Peridium* composed of several layers of hyaline to brown cells of *textura angularis*. *Hamathecium* hyphae-like, pseudoparaphyses, embedded in a gelatinous matrix. *Asci* 8-spored, bitunicate, fissitunicate, broad clavate to cylindric-clavate, short pedicellate. *Ascospores* biseriate, fusiform with obtuse ends, hyaline, 1-septate, guttulate, smooth-walled, surrounded by a mucilaginous sheath. **Asexual morph**: Undetermined.

*Notes*: The monotypic genus was introduced to accommodate *Neoangustimassarina sichuanensis*, which formed a distinct clade within Amorosiaceae (Figure 1). *Neoangustimassarina* resembles *Angustimassarina* in forming globose to subglobose ascomata, hyaline, and septate ascospores surrounded by mucilaginous sheaths [38]. However, *Neoangustimassarina* differs from the latter in having immersed ascomata without a pore opening, broader asci (broad clavate to cylindric-clavate vs. cylindrical to cylindric-clavate), and the septa of the ascospores (1-septate vs. 1–3-septate). We, hereby, introduce the new genus based on the distinctiveness of morphology and multi-gene phylogeny.

***Neoangustimassarina sichuanensis*** X.D. Yu, S.N. Zhang and Jian K. Liu, sp. nov., Figure 2

*MycoBank*: MB 843717

*Etymology*: The epithet refers to Sichuan Province where the fungus was collected.

*Holotype*: HKAS 123092

*Saprobic* on dead wood in terrestrial habitat. **Sexual morph**: *Ascomata* solitary, scattered, immersed, visible as circular, pale brown to nearly white flat cap, with a small black dot in the center, in vertical section 135–235 μm high, 190–260 μm diam., subglobose, uniloculate, ostiolate. *Peridium* 10–23 μm wide, composed of several layers of hyaline to brown cells of *textura angularis*. *Hamathecium* 1.9–2.9 µm wide, hyphae-like, pseudoparaphyses, embedded in a gelatinous matrix. *Asci* 78–125 × 20–30 µm ($\bar{x}$ = 92 × 25 µm, *n* = 30), 8-spored, bitunicate, fissitunicate, broad clavate to cylindric-clavate, straight or slightly curved, short pedicellate to subsessile, rounded at the apex, with an ocular chamber. *Ascospores* 23–35 × 6.5–10.5 µm ($\bar{x}$ = 30 × 9 µm, *n* = 30), overlapping biseriate, fusiform with obtuse ends, hyaline, 1-septate, constricted at the septum, the upper cell slightly wider than the lower cell, guttulate when young, smooth-walled, surrounded by a wide mucilaginous sheath. **Asexual morph**: Undetermined.

*Culture characteristics*: Colonies on PDA reaching 50–60 mm after 7 weeks at 25 °C, circular, dry, the mycelium sparse at the margin, greyish-brown, reverse dark brown.

*Material examined*: CHINA, Sichuan Province, Chengdu City, Pengzhou County, Huilonggou Scenic Area, 31°14′21″ N, 103°47′28″ E, 1135 m Elevation, on dead wood, 28 July 2021, X.D. Yu, HLG3 (HKAS 123092, holotype); ex-holotype living culture CGMCC 3.20937; *ibid*., HUEST 22.0001, isotype, ex-isotype living culture UESTCC 22.0001.

**Bambusicolaceae** D.Q. Dai and K.D. Hyde, *Fungal Diversity* 63: 49 (2013)

*Notes*: Bambusicolaceae was established by Hyde et al. [13] to accommodate *Bambusicola*, D.Q. Dai and K.D. Hyde [43]. Four genera were accepted in this family, *viz. Bambusicola* [43], *Corylicola* [44], *Leucaenicola* [41] and *Palmiascoma* [45]. Most *Bambusicola* species are parasites or saprobes and have been found on Bamboos (Poaceae) in terrestrial habitats [43,46,47,48,49,50], except *B. aquatica* (from a freshwater habitat) and *B. ficuum* (on *Ficus* sp., Moraceae) [51,52]. In this study, a coelomycetous *Bambusicola* species is introduced.

***Bambusicola*** D.Q. Dai and K.D. Hyde, *Cryptogamie Mycologie* 33: 367 (2012)

***Bambusicola guttulata*** X.D. Yu, S.N. Zhang and Jian K. Liu, sp. nov., Figure 3.

*MycoBank*: MB 843718

*Etymology*: Referring to the conidia with large guttules.

*Holotype*: HKAS 123091

*Saprobic* on dead branches of bamboo. **Sexual morph**: Undetermined. **Asexual morph**: *Conidiomata* 100–170 μm high, 130–250 μm diam., dark brown to black, pycnidial, usually forming in a linear series on the host surface, solitary, closed when young, becoming erumpent, stromatic, irregular subglobose in section, immersed or semi-immersed, unilocular, thick-walled. *Conidiomatal wall* 25–55 μm wide, composed of thick-walled, subhyaline to brown cells of *textura angularis*. *Conidiophores* hyaline, cylindrical, branched, straight or slightly flexuous, septate, and occasionally reduced to conidiogenous cells. *Conidiogenous cells* 6–16 × 3–5 μm, holoblastic, hyaline, cylindrical to subcylindrical, determinate, terminal, smooth-walled. *Conidia* 14–21 × 4–6 μm ($\bar{x}$ = 17 × 5 μm; *n* = 30), hyaline to pale brown, unicellular when young, becoming 1-septate at maturity, cylindrical to subcylindrical, sometimes with a narrow and truncate base, straight or slightly curved, smooth-walled, guttulate.

*Culture characteristics*: Colonies on PDA reaching 30–40 mm after 7 weeks at 25 °C, irregular, raised to umbonate, surface rough, dense, edge undulate, greyish-yellow, dry, reverse dark brown.

*Material examined*: CHINA, Sichuan Province, Chengdu City, Tianfu New Area, Dalin Village, 30°16′43″ N, 104°6′44″ E, 500 m Elevation, on dead branches of bamboo, 24 July 2021, X.D. Yu, B2 (HKAS 123091, holotype); ex-holotype living culture CGMCC 3.20935; *ibid*., HUEST 22.0002, isotype, ex-isotype living culture UESTCC 22.0002.

*Notes*: Two isolates of *Bambusicola guttulata* formed a distinct lineage in *Bambusicola* (Figure 1). Morphologically, *B. guttulata* is most similar to the asexual morph of the generic type *B. massarinia* compared to the anamorphic species in the genus *Bambusicola* [43]. However, *B. guttulata* has broader conidia than that of *B. massarinia* (14–21 × 4–6 μm vs. 14–20 × 2–3 μm). The establishment of the new species *B. guttulata* is justified by morphological and phylogenetic evidence.

**Lophiostomataceae** Sacc., *Sylloge Fungorum* 2: 672 (1883)

*Notes*: Nitschke [53] introduced “Lophiostomeae” based on the type species of *Lophiostoma*
*macrostomum* (Tode) Ces. and De Not. Saccardo formally established the family Lophiostomataceae and placed “Lophiostomeae” in the order Pleosporales [54]. Members of this family have crest-like ostioles in most cases and easily to be recognized. They are characterized by immersed to erumpent ascomata, mostly clavate asci, hyaline to dark brown ascospores with appendages or mucilaginous sheaths [38,55]. With the continuous increase of new members, the family currently comprises 30 genera [16]. A new species added to the genus *Flabellascoma* is identified and described.

***Flabellascoma*** A. Hashim., K. Hiray. and Kaz. Tanaka, *Studies in Mycology* 90: 167 (2018)

***Flabellascoma sichuanense*** X.D. Yu, S.N. Zhang and Jian K. Liu, sp. nov., Figure 4.

*MycoBank*: MB 843719

*Etymology*: The epithet refers to Sichuan Province where the fungus was collected.

*Holotype*: HKAS 123094

*Saprobic* on dead branches of *Eriobotrya* sp. (Rosaceae). **Sexual morph**: *Ascomata* solitary, scattered, rarely clustered, immersed to erumpent, visible as black, crest-like ostiolar neck on the substrate, in vertical section 150–350 μm high, 190–280 μm diam., subglobose, uniloculate. *Ostiole* central, laterally compressed, periphysate. *Peridium* 25–35 μm wide, composed of several layers of brown, thick-walled cells of *textura angularis*. *Hamathecium* 1.5–3.5 µm wide, hyphae-like, pseudoparaphyses, embedded in a gelatinous matrix. *Asci* 55–75 × 9.5–13 µm ($\bar{x}$ = 64 × 11 µm, *n* = 30), 8-spored, bitunicate, fissitunicate, cylindric-clavate, straight or slightly curved, shortly pedicellate, rounded at the apex, with an ocular chamber. *Ascospores* 15–18 × 5–7 µm ($\bar{x}$ = 16.5 × 5.5 µm, *n* = 30), overlapping biseriate, fusiform, hyaline, 1-septate, constricted at the septum, the upper cell slightly wider than the lower cell, guttulate, smooth-walled, with a narrow bipolar sheath. *Sheath* 3.0–7.0 μm long, 1.5–2.5 μm wide, drawn-out at both ends, with an internal chamber at both ends of ascospores (Figure 4l). **Asexual morph**: Undetermined.

*Culture characteristics*: Colonies on PDA reaching 40–50 mm after 7 weeks at 25 °C, circular, with dense mycelium on the surface, dark grayish of the inner ring, and brown of the outer ring; in reverse black of the inner ring, and brown of the outer ring.

*Material examined*: CHINA, Sichuan Province, Chengdu City, Tianfu New Area, Dalin Village, 30°16′43″ N, 104°6′44″ E, 500 m Elevation, on dead branches of *Eriobotrya* sp. (Rosaceae), 24 July 2021, X.D. Yu, L4 (HKAS 123094, holotype); ex-holotype living culture CGMCC 3.20936; *ibid.*, HUEST 22.0003, isotype, ex-isotype living culture UESTCC 22.0003.

*Notes*: The phylogenetic result based on SSU, ITS, LSU, *RPB2* and *TEF1α* sequence data showed that the new collection *Flabellascoma sichuanense* nested in *Flabellascoma* (Figure 1) and formed a distinct lineage. Morphologically, it fits well with the genus *Flabellascoma* in having immersed ascomata, bitunicate, fissitunicate, cylindrical, clavate asci and fusiform, hyaline, 1-septate ascospores with a narrow bipolar sheath [56]. However, the dimensions of asci and ascospores distinguish *F. sichuanense* from other species (Table 3).

**Occultibambusaceae** D.Q. Dai and K.D. Hyde, *Fungal Diversity* 82: 25 (2017)

*Notes*: Species of Occultibambusaceae are mostly saprobic and frequently found on monocotyledons or hardwood trees in terrestrial and aquatic habitats [47,58]. Dai et al. [47] established this family to accommodate *Neooccultibambusa*, *Occultibambusa*, *Seriascoma* and *Versicolorisporium*. *Brunneofusispora*, typified by *Brunneofusispora sinensis*, was subsequently introduced to this family by Phookamsak et al. [59]. Phylogenetically, the coelomycetous genus *Versicolorisporium* appeared to be a close relationship with *Occultibambusa* in previous studies [21,52,60,61]. However, they continue to be treated as two distinct genera because the known asexual morph of *Occultibambusa* is different from *Versicolorisporium* [60]. In this study, a new *Occultibambusa* species is introduced.

***Occultibambusa*** D.Q. Dai and K.D. Hyde, *Fungal Diversity* 82: 25 (2017)

***Occultibambusa sichuanensis*** X.D. Yu, S.N. Zhang and Jian K. Liu, sp. nov., Figure 5.

*MycoBank*: MB 843720

*Etymology*: The epithet refers to Sichuan Province where the fungus was collected.

*Holotype*: HKAS 123093

*Saprobic* on dead branches of Bamboo. **Sexual morph**: *Ascomata* solitary to gregarious, semi-immersed, visible as black domes on the substrate, in vertical section 130–180 μm high, 340–440 μm diam., subglobose, coriaceous, uniloculate. *Peridium* 25–90 μm wide, composed of several layers of brown, thick-walled cells of *textura angularis*. *Hamathecium* 2.8–3.3 µm wide, hyphae-like, cellular pseudoparaphyses, embedded in a gelatinous matrix. *Asci* 70–100 × 22–27 µm ($\bar{x}$ = 85 × 24 µm, *n* = 30), 8-spored, bitunicate, fissitunicate, obovoid to pyriform, straight or slightly curved, shortly pedicellate, rounded at the apex, with an ocular chamber. *Ascospores* 31–41 × 6.5–10 µm ($\bar{x}$ = 36 × 8 µm, *n* = 30), 3-seriate, fusiform, straight to somewhat curved, brown, 1-septate, constricted at the septum, guttulate, smooth-walled, surrounded by a mucilaginous sheath. **Asexual morph**: Undetermined.

*Culture characteristics*: Colonies on PDA reaching 40–50 mm after 7 weeks at 25 °C, circular, with sparse mycelium on the surface, light gray of the inner ring, and brown of the outer ring; in reverse olive green.

*Material examined*: CHINA, Sichuan Province, Chengdu City, Pengzhou County, Huilonggou Scenic Area, 31°14′21″ N, 103°47′28″ E, 1135 m Elevation, on dead branches of bamboo in a terrestrial environment, 28 July 2021, X.D. Yu, HLG8 (HKAS 123093, holotype); ex-holotype living culture CGMCC 3.20938; *ibid*., HUEST 22.0004, isotype, ex-isotype living culture UESTCC 22.0004.

*Notes*: The blast search based on LSU sequence data of our new collection showed that the closest hits were *Versicolorisporium triseptatum* (HHUF 28815 = JCM14775, identity 99.18%; NMX1222, identity 99.03%), and *Occultibambusa bambusae* (MFLUCC 11-0394, identity 98.01%); the closest hits based on ITS sequence were *Versicolorisporium triseptatum* (JCM 14775, identity 93.95%; NMX1222, identity 93.72%), and *Occultibambusa hongheensis* (KUMCC 21-0020, identity 90.95%); the closest hits based on *TEF1**α* sequence were *Occultibambusa hongheensis* (KUMCC 21-0020, identity 97.02%), and *O. maolanensis* (KUMCC 21-0020, identity 96.91%). Multi-gene phylogeny showed that the new collection grouped with *Occultibambusa* and *Versicolorisporium*. It formed a sister clade with *V. triseptatum* with high statistical support (100% ML/1.00 BYPP, Figure 1). However, the morphology of our collection fits well with *Occultibambusa*. Further morphological evidence of its association with *Versicolorisporium* is somewhat difficult due to the lack of asexual morph in our collection. Therefore, we recognize our new collection as a new species of *Occultibambusa*, namely, *O. sichuanensis.* The morphological comparison of *Occultibambusa* species was listed in Table 4.

**Tetraplosphaeriaceae** Kaz. Tanaka and K. Hiray, *Studies in Mycology* 64: 177 (2009)

*Notes*: Tetraplosphaeriaceae was introduced by Tanaka et al. [64], and typified by *Tetraplosphaeria*. The latest taxonomic treatment of the family contains nine genera [1]. *Pseudotetraploa* is a genus with only known asexual forms, which were commonly associated with Poaceae (*Dendrocalamus stocksii*, *Pleioblastus chino*, *Pleioblastus chino*, *Sasa kurilensis*) distributed in Japan or India [64,65]. In this study, a new *Pseudotetraploa* species associated with bamboos from China is introduced.

***Pseudotetraploa*** Kaz. Tanaka and K. Hiray, *Studies in Mycology* 64: 193 (2009)

***Pseudotetraploa bambusicola*** X.D. Yu, S.N. Zhang and Jian K. Liu, sp. nov., Figure 6.

*MycoBank*: MB 843721

*Etymology*: Refers to the bamboo host.

*Holotype*: HKAS 123095

*Saprobic* on dead branches of Bamboo. **Sexual morph**: Undetermined. **Asexual morph**: *Colonies* effuse, black. *Mycelium* superficial. *Conidiophores* absent. *Conidiogenous cells* monoblastic, integrated, usually indistinguishable from superficial hyphae. *Conidia* 23–41 × 14–24 µm ($\bar{x}$ = 32 × 19 µm, *n* = 50), solitary, amygdaliform to ovoid, or obovoid, dark brown to black, pseudoseptate, consisting of 3–4 columns, with 0–4 setose appendages. *Appendages* 8.85–95 × 2.5–4.5 µm ($\bar{x}$ = 36 × 3.5 µm, *n* = 50), 0–4-septate, dark brown, smooth, unbranched, straight or curved.

*Culture characteristics*: Colonies on PDA reaching 40–50 mm after 7 weeks at 25 °C, circular, dry, with dense mycelium, raised, entire at the edge, grayish brown, reverse dark brown.

*Material examined*: CHINA, Sichuan Province, Chengdu City, Longquanyi District, Longquan Mountain Scenic Area, 30°32′47″ N, 104°19′11″ E, 800 m Elevation, on dead branches of bamboo in a terrestrial environment, 13 August 2021, X.D. Yu, THGL14 (HKAS 123095, holotype); ex-holotype living culture CGMCC 3.20939; *ibid.*, HUEST 22.0005, isotype, ex-isotype living culture UESTCC 22.0005.

Notes: The phylogenetic result (Figure 1) showed that *Pseudotetraploa bambusicola* formed a distinct lineage within *Pseudotetraploa* [64]. Morphologically, *Pseudotetraploa bambusicola* resembles *P. curviappendiculata*, *P. javanica*, *P. longissimi* and *P. rajmachiensis* in having monoblastic conidiogenous cells. However, they can be distinguished by the shape of conidia (obclavate to narrowly obpyriform in *P. curviappendiculata* and *P. longissimi*; ovoid in *P. javanica*, ovoid to obclavate or obpyriform in P. rajmachiensis; amygdaliform to ovoid, or obovoid in *P. bambusicola*) [64].

## 4. Discussion

The genus *Flabellascoma* was introduced by Hashimoto et al. [56] to accommodate two terrestrial species *F. cycadicola* A. Hashim., K. Hiray and Kaz. Tanaka and *F. minimum*. Subsequently, two species *F. aquaticum* D.F. Bao, Z.L. Luo, K.D. Hyde and H.Y. Su and *F. fusiforme* D.F. Bao, Z.L. Luo, K.D. Hyde and H.Y. Su from freshwater habitats were introduced based on multi-gene phylogeny [57]. Members of *Flabellascoma* have similar morphological features [56,57] and it is difficult to distinguish *Flabellascoma* species by the size and shape of asci and ascospores [57]. Bao et al. [57] proposed that the ascomatal features appear to be remarkable features to distinguish taxa in this genus. Molecular data were found to be more supportive for the identification of the new species in this study, and we believed that DNA data provided more objective evidence for the species distinction of *Flabellascoma*.

In previous studies, the relationship between *Occultibambusa* and *Versicolorisporium* has not been well resolved due to the asexual morphs of *Occultibambusa* and *Versicolorisporium* being inconsistent [60]. In our phylogenetic tree, however, the genus *Occultibambusa* is not monophyletic (Figure 1), of which *O. fusispora* formed an independent lineage in Occultibambusaceae; this is consistent with recent relevant studies [60,61]. *Occultibambusa fusispora* is the only species in the genus reported with its holomorph [47]; we cannot solve the problem between *Occultibambusa* and *Versicolorisporium* due to the type-of-species issue, although *O. fusispora* has an asexual morph. Therefore, further studies are needed to provide sexual and asexual links of the type of species of *O. bambusae* and *V. triseptatum* towards the classification of *Occultibambusa* and *Versicolorisporium* with more sampling and taxa population included in the analysis.

During the investigation of microfungi in Sichuan Province, we randomly sampled three times in the vicinity of Chengdu city from July to August 2021. Morphological and phylogenetic results showed that these newly collected interesting taxa were distributed in five different pleosporalean families. It is worth noting that three new species found on bamboo are typical bambusicolous fungi. Bamboo is a gramineous plant with economic and ornamental value, and its culms and leaves are abundant in saprobic fungi [44,66,67,68]. China has the richest bamboo resources, with a total of 861 species distributes in 43 genera [69]. Among them, Sichuan has a large area of bamboo forests, with an area of 592,800 ha, ranking fifth in the country after Fujian, Jiangxi, Zhejiang and Hunan [69]. In recent years, an increasing number of new species of bambusicolous fungi have been reported and discovered in China [60,67,70,71]. Therefore, the unique natural conditions in Sichuan are of great potential for the excavation and identification of bamboo fungi.

## Figures and Tables

**Figure 1 jof-08-00720-f001:**
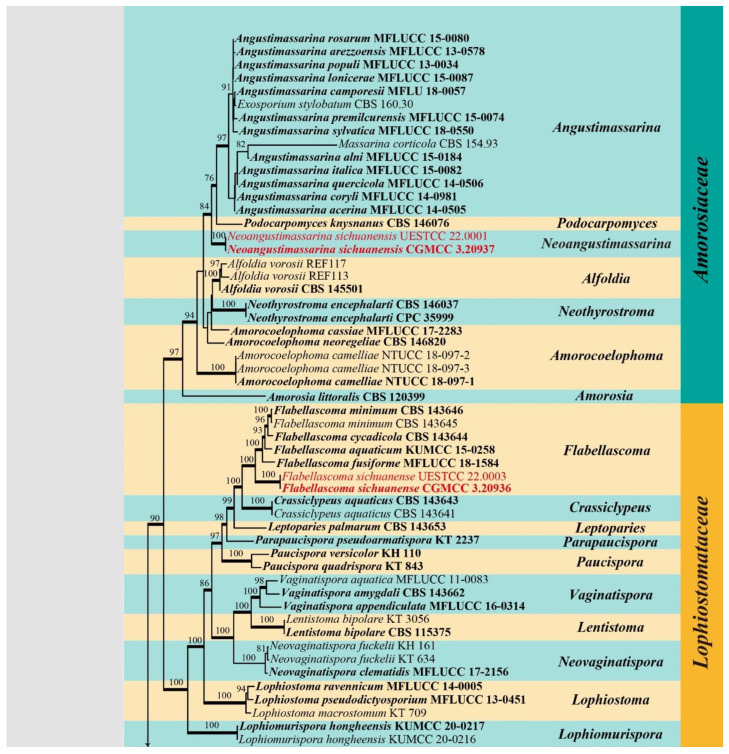
RAxML tree generated from combined SSU, ITS, LSU, *RPB2* and *TEF1α* sequence data of targeted five families (Amorosiaceae, Bambusicolaceae, Lophiostomataceae, Occultibambusaceae, and Tetraplosphaeriaceae) in Pleosporales. Bootstrap values for ML equal to or greater than 75% are placed above the branches. Branches with Bayesian posterior probabilities (PP) from MCMC analysis equal to or greater than 0.95 are in bold. The tree was rooted with *Hysterium rhizophorae* (NFCCI 4250). The ex-type strains are indicated in bold and newly generated sequences are indicated in red.

**Figure 2 jof-08-00720-f002:**
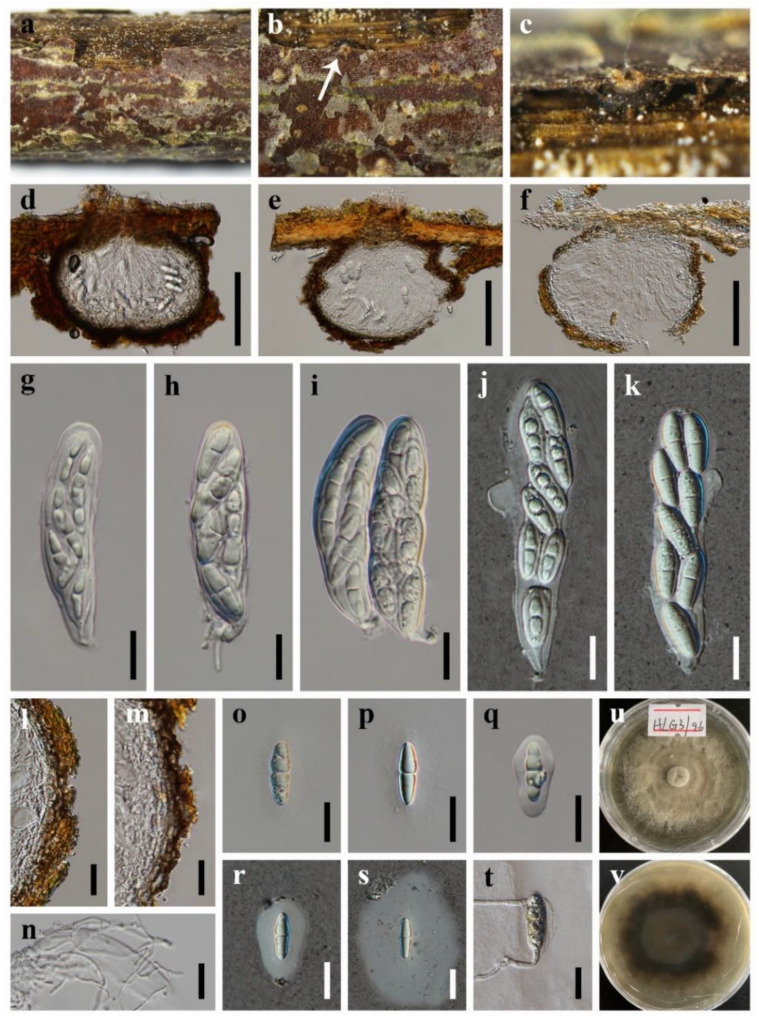
***Neoangustimassarina sichuanensis*** (HKAS 123092, **holotype**) (**a**–**c**) Ascomata on host substrate. (**d**–**f**) Vertical section of ascoma. (**g**–**k**) Asci. (**l**,**m**) Structure of peridium. (**n**) Pseudoparaphyses. (**o**–**s**) Ascospores. (**t**) Germinated ascospore. (**u**,**v**) Colonies on PDA, above (**u**) and below (**v**). Scale bars: (**d**–**f**) = 100 μm, (**g**–**t**) = 20 μm.

**Figure 3 jof-08-00720-f003:**
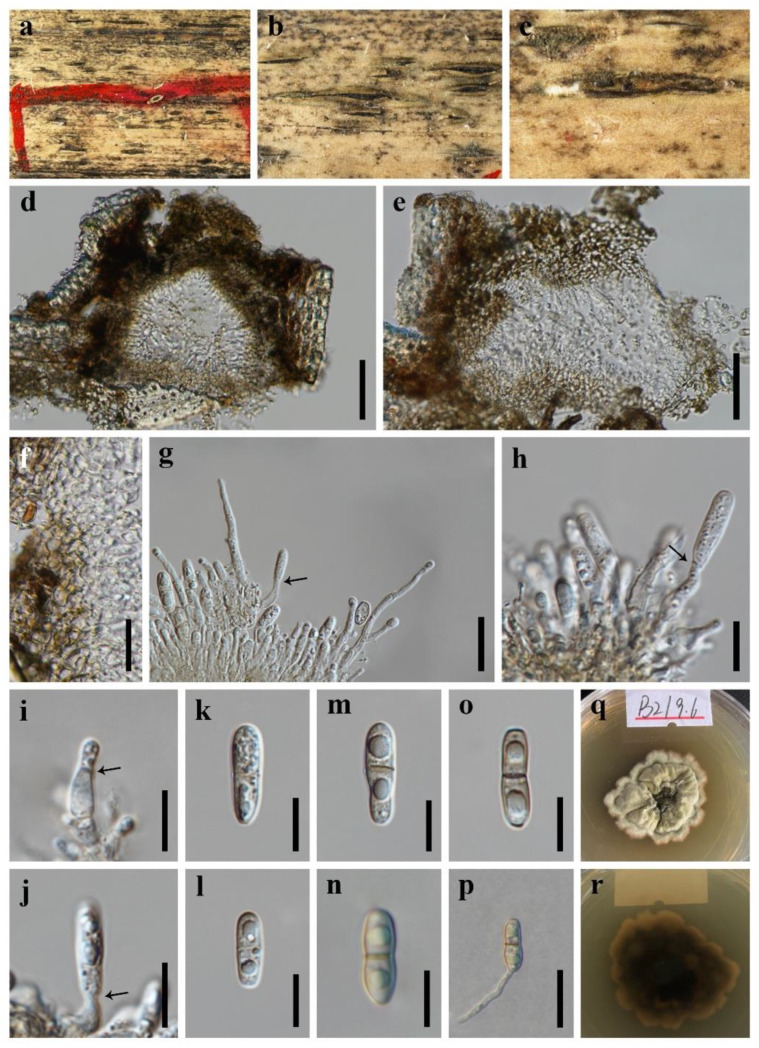
***Bambusicola guttulata*** (HKAS 123091, **holotype**) (**a**–**c**) Conidiomata on surface of dead bamboo culms. (**d**,**e**) Vertical section of conidioma. (**f**) Wall of conidioma. (**g**–**j**) Conidiogenous cells bearing conidia (the arrows indicated how the conidiogenous cells produce conidia). (**k**–**o**) Conidia. (**p**) Germinating conidia. (**q**,**r**) Colonies on PDA, above and below. Scale bars: (**d**,**e**) = 50 μm, (**f**,**g**,**p**) = 20 μm, (**h**–**o**) = 10 μm.

**Figure 4 jof-08-00720-f004:**
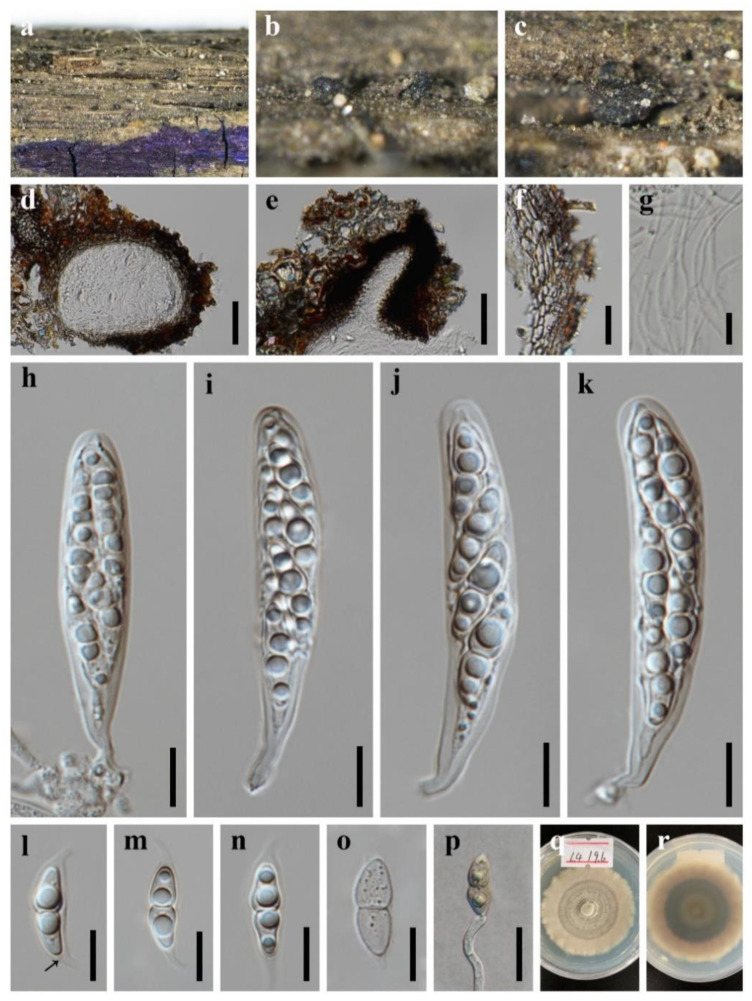
***Flabellascoma sichuanense*** (HKAS 123094, **holotype**) (**a**–**c**) Ascomata on host substrate. (**d**) Vertical section of ascoma. (**e**) Ostiole, showing periphyses. (**f**) Structure of peridium. (**g**) Pseudoparaphyses. (**h**–**k**) Asci. (**l**–**o**) Ascospores. (**p**) Germinated ascospore. (**q**,**r**) Colonies on PDA, above and below. Scale bars: (**d**,**e**) = 50 μm, (**f**,**p**) = 20 μm, (**g**–**o**) = 10 μm.

**Figure 5 jof-08-00720-f005:**
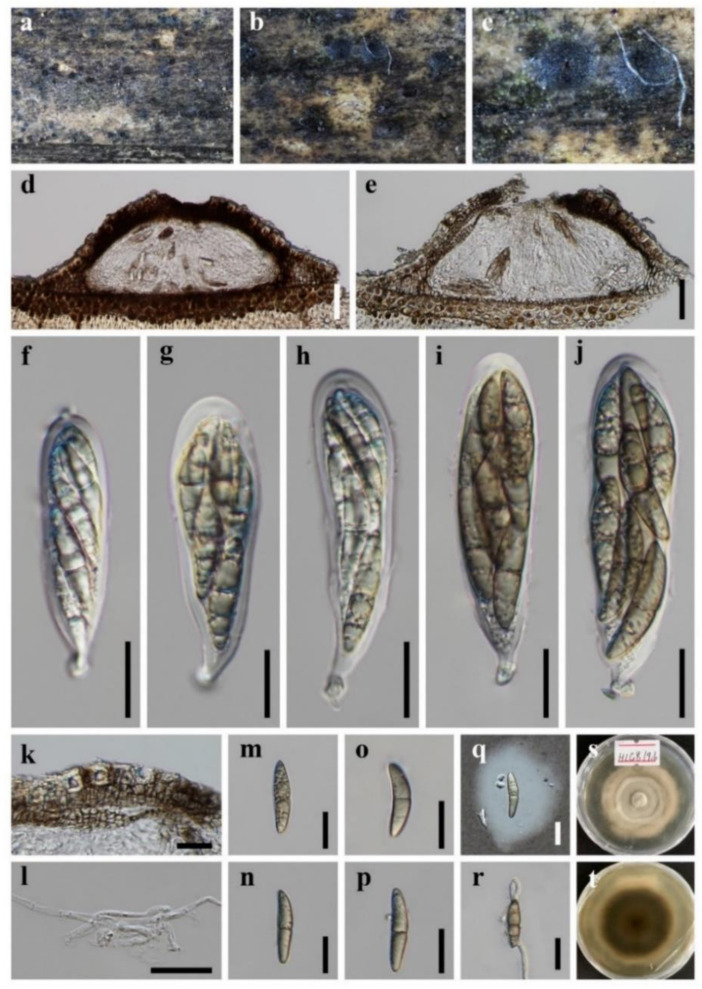
***Occultibambusa sichuanensis*** (HKAS 123093, **holotype**) (**a**–**c**) Ascomata on host substrate. (**d**,**e**) Vertical section of ascoma. (**f**–**j**) Asci. (**k**) Structure of peridium. (**l**) Pseudoparaphyses. (**s**) Ascospores. (**t**) Germinated ascospore. Colonies on PDA, above and below. Scale bars: (**d**,**e**) = 50 μm, (**f**–**k**,**m**–**t**) = 20 μm, (**l**) = 10 μm.

**Figure 6 jof-08-00720-f006:**
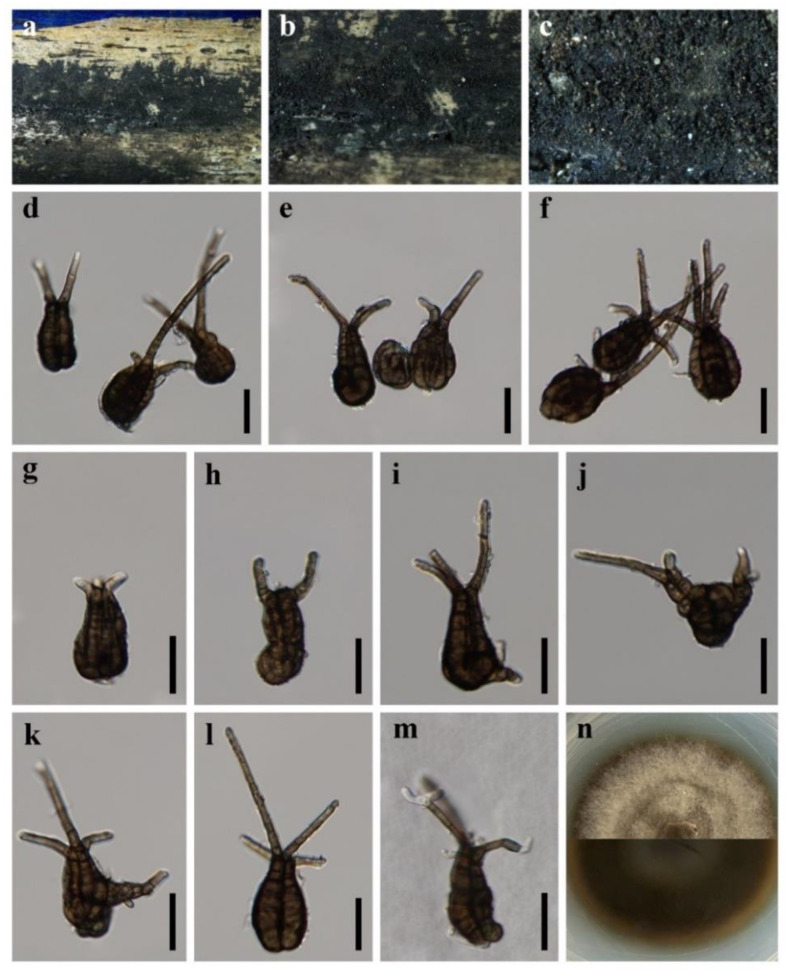
***Pseudotetraploa bambusicola*** (HKAS 123095, **holotype**) (**a**–**c**) Colonies on natural substratum. (**d**–**l**) Conidia. (**m**) Germinating conidium. (**n**) Colony on PDA from above and below. Scale bars: (**d**–**m**) = 20 μm.

**Table 1 jof-08-00720-t001:** PCR thermal cycles for SSU, ITS, LSU, *RPB2* and *TEF1α* amplification.

Step	SSU			ITS, LSU, *RPB2*			*TEF1α*		
	Temperature	Time	Cycles	Temperature	Time	Cycles	Temperature	Time	Cycles
Initial Denaturation	98 °C	2 min	1	98 °C	2 min	1	98 °C	2 min	1
Denaturation	98 °C	10 s	35	98 °C	10 s	35	98 °C	10 s	35
Annealing	47 °C	10 s		56 °C	10 s		61.7 °C	10 s	
Extension	72 °C	10 s		72 °C	10 s		72 °C	10 s	
Final Extension	72 °C	5 min	1	72 °C	5 min	1	72 °C	5 min	1
Hold	4 °C	-	-	4 °C	-	-	4 °C	-	-

**Table 2 jof-08-00720-t002:** Taxa used in the phylogenetic analyses and their GenBank accession numbers. Newly generated sequences are indicated with * and the ex-type strains are in bold.

Species	Voucher/Strain/Isolate	GenBank Accession Number				
		SSU	ITS	LSU	*RPB2*	*TEF1α*
** *Alfoldia vorosii* **	**CBS 145501**	**MK589346**	**JN859336**	**MK589354**	**N/A**	**MK599320**
*Alfoldia vorosii*	REF113	MK589345	JN859333	MK589353	N/A	MK599319
*Alfoldia vorosii*	REF117	MK589347	JN859337	MK589355	N/A	MK599321
** *Amorocoelophoma camelliae* **	**NTUCC 18-097-1**	**MT071230**	**MT112303**	**MT071279**	**MT459143**	**MT743271**
*Amorocoelophoma camelliae*	NTUCC 18-097-2	MT071231	MT112304	MT071280	MT459141	MT743272
*Amorocoelophoma camelliae*	NTUCC 18-097-3	MT071232	MT112305	MT071281	MT459142	MT743273
** *Amorocoelophoma cassiae* **	**MFLUCC 17-2283**	**MK347847**	**MK347739**	**MK347956**	**MK434894**	**MK360041**
** *Amorocoelophoma neoregeliae* **	**CBS 146820**	**N/A**	**MZ064410**	**MZ064467**	**MZ078193**	**MZ078247**
** *Amorosia littoralis* **	**CBS 120399**	**AM292056**	**AM292047**	**AM292055**	**N/A**	**N/A**
** *Angustimassarina acerina* **	**MFLUCC 14-0505**	**KP899123**	**KP899132**	**KP888637**	**N/A**	**KR075168**
** *Angustimassarina arezzoensis* **	**MFLUCC 13-0578**	**KY501113**	**KY496743**	**KY496722**	**N/A**	**KY514392**
** *Angustimassarina camporesii* **	**MFLU 18-0057**	**MN244173**	**MN244197**	**MN244167**	**N/A**	**N/A**
** *Angustimassarina italica* **	**MFLUCC 15-0082**	**KY501124**	**KY496756**	**KY496736**	**N/A**	**KY514400**
** *Angustimassarina lonicerae* **	**MFLUCC 15-0087**	**N/A**	**KY496759**	**KY496724**	**N/A**	**N/A**
** *Angustimassarina populi* **	**MFLUCC 13-0034**	**KP899128**	**KP899137**	**KP888642**	**N/A**	**KR075164**
** *Angustimassarina premilcurensis* **	**MFLUCC 15-0074**	**N/A**	**KY496745**	**KY496725**	**KY514404**	**N/A**
** *Angustimassarina quercicola* **	**MFLUCC 14-0506**	**KP899124**	**KP899133**	**KP888638**	**N/A**	**KR075169**
** *Angustimassarina rosarum* **	**MFLUCC 15-0080**	**N/A**	**MG828869**	**MG828985**	**N/A**	**N/A**
** *Angustimassarina sylvatica* **	**MFLUCC 18-0550**	**MK314097**	**MK307843**	**MK307844**	**N/A**	**MK360181**
** *Angustimassarina alni* **	**MFLUCC 15-0184**	**KY548098**	**KY548099**	**KY548097**	**N/A**	**N/A**
** *Angustimassarina coryli* **	**MFLUCC 14-0981**	**N/A**	**MF167431**	**MF167432**	**N/A**	**MF167433**
** *Aquatisphaeria thailandica* **	**MFLUCC 21-0025**	**MW890967**	**MW890969**	**MW890763**	**N/A**	**N/A**
** *Bambusicola aquatica* **	**MFLUCC 18-1031**	**MT864293**	**MT627729**	**MN913710**	**MT878462**	**MT954392**
** *Bambusicola bambusae* **	**MFLUCC 11-0614**	**JX442039**	**JX442031**	**JX442035**	**KP761718**	**KP761722**
** *Bambusicola didymospora* **	**MFLUCC 10-0557**	**KU872110**	**KU940116**	**KU863105**	**KU940163**	**KU940188**
** *Bambusicola dimorpha* **	**MFLUCC 13-0282**	**KY038354**	**KY026582**	**KY000661**	**KY056663**	**N/A**
** *Bambusicola ficuum* **	**MFLUCC 17-0872**	**MT215581**	**N/A**	**MT215580**	**N/A**	**MT199326**
** *Bambusicola fusispora* **	**MFLUCC 20-0149**	**MW076529**	**MW076532**	**MW076531**	**MW034589**	**N/A**
***Bambusicola guttulata* ***	**CGMCC 3.20935**	**ON332919**	**ON332909**	**ON332927**	**ON383985**	**ON381177**
*Bambusicola guttulata* *	UESTCC 22.0002	ON332920	ON332910	ON332928	ON383986	ON381178
** *Bambusicola irregulispora* **	**MFLUCC 11-0437**	**JX442040**	**JX442032**	**JX442036**	**KP761719**	**KP761723**
** *Bambusicola loculata* **	**MFLUCC 13-0856**	**KP761735**	**KP761732**	**KP761729**	**KP761715**	**KP761724**
** *Bambusicola massarinia* **	**MFLUCC 11-0389**	**JX442041**	**JX442033**	**JX442037**	**KP761716**	**KP761725**
** *Bambusicola pustulata* **	**MFLUCC 15-0190**	**KU872112**	**KU940118**	**KU863107**	**KU940165**	**KU940190**
** *Bambusicola sichuanensis* **	**SICAUCC 16-0002**	**MK253528**	**MK253473**	**MK253532**	**MK262830**	**MK262828**
** *Bambusicola splendida* **	**MFLUCC 11-0439**	**JX442042**	**JX442034**	**JX442038**	**KP761717**	**KP761726**
** *Bambusicola subthailandica* **	**SICAU 16-0005**	**MK253529**	**MK253474**	**MK253533**	**MK262831**	**MK262829**
** *Bambusicola thailandica* **	**MFLUCC 11-0147**	**N/A**	**KU940119**	**KU863108**	**KU940166**	**KU940191**
** *Bambusicola triseptatispora* **	**MFLUCC 11-0166**	**N/A**	**KU940120**	**KU863109**	**KU940167**	**N/A**
** *Brunneofusispora clematidis* **	**MFLUCC 17-2070**	**MT226685**	**MT310615**	**MT214570**	**MT394692**	**MT394629**
** *Brunneofusispora inclinatiostiola* **	**CGMCC 3.20403**	**MZ964884**	**MZ964866**	**MZ964875**	**OK061075**	**OK061069**
** *Brunneofusispora sinensis* **	**KUMCC 17-0030**	**MH393556**	**MH393558**	**MH393557**	**N/A**	**MH395329**
*Corylicola italica*	MFLU 19-0500	MT554923	MT554925	MT554926	MT590776	**N/A**
** *Corylicola italica* **	**MFLUCC 20-0111**	**MT633084**	**MT633085**	**MT626713**	**MT635596**	**MT590777**
*Crassiclypeus aquaticus*	CBS 143641	LC312470	LC312499	LC312528	LC312586	LC312557
** *Crassiclypeus aquaticus* **	**CBS 143643**	**LC312472**	**LC312501**	**LC312530**	**LC312588**	**LC312559**
** *Ernakulamia krabiensis* **	**MFLUCC 18–0237**	**MK347880**	**MK347773**	**MK347990**	**N/A**	**N/A**
** *Ernakulamia tanakae* **	**NFCCI 4615**	**N/A**	**MN937229**	**MN937211**	**N/A**	**N/A**
** *Ernakulamia xishuangbannaensis* **	**KUMCC 17-0187**	**MH260354**	**MH275080**	**MH260314**	**N/A**	**N/A**
*Exosporium stylobatum*	CBS 160.30	N/A	JQ044428	JQ044447	N/A	N/A
** *Flabellascoma aquaticum* **	**KUMCC 15-0258**	**MN304832**	**MN304827**	**MN274564**	**MN328895**	**MN328898**
** *Flabellascoma cycadicola* **	**CBS 143644**	**LC312473**	**LC312502**	**LC312531**	**LC312589**	**LC312560**
** *Flabellascoma fusiforme* **	**MFLUCC 18-1584**	**N/A**	**MN304830**	**MN274567**	**N/A**	**MN328902**
*Flabellascoma minimum*	CBS 143645	LC312474	LC312503	LC312532	LC312590	LC312561
** *Flabellascoma minimum* **	**CBS 143646**	**LC312475**	**LC312504**	**LC312533**	**LC312591**	**LC312562**
***Flabellascoma sichuanense* ***	**CGMCC 3.20936**	**ON332921**	**ON332911**	**ON332929**	**ON383987**	**ON381179**
*Flabellascoma sichuanense* *	UESTCC 22.0003	ON332922	ON332912	ON332930	ON383988	ON381180
*Hysterium rhizophorae*	NFCCI-4250	MG844280	MG844284	MG844276	MG968956	N/A
** *Lentistoma bipolare* **	**CBS 115375**	**LC312477**	**LC312506**	**LC312535**	**LC312593**	**LC312564**
*Lentistoma bipolare*	KT 3056	LC312484	LC312513	LC312542	LC312600	LC312571
** *Leptoparies palmarum* **	**CBS 143653**	**LC312485**	**LC312514**	**LC312543**	**LC312601**	**LC312572**
** *Leucaenicola aseptata* **	**MFLUCC 17-2423**	**MK347853**	**MK347746**	**MK347963**	**MK434891**	**MK360059**
** *Leucaenicola camelliae* **	**NTUCC 18-093-4**	**MT071229**	**MT112302**	**MT071278**	**MT743283**	**MT374091**
** *Leucaenicola phraeana* **	**MFLUCC 18-0472**	**MK347892**	**MK347785**	**MK348003**	**MK434867**	**MK360060**
*Lophiomurispora hongheensis*	KUMCC 20-0216	MW264227	MW264218	MW264197	MW256810	MW256819
** *Lophiomurispora hongheensis* **	**KUMCC 20-0217**	**MW264225**	**MW264216**	**MW264195**	**MW256808**	**MW256817**
*Lophiostoma macrostomum*	KT 709/HHUF 27293	AB521732	AB433276	AB433274	JN993493	LC001753
** *Lophiostoma pseudodictyosporium* **	**MFLUCC 13-0451**	**N/A**	**KR025858**	**KR025862**	**N/A**	**N/A**
** *Lophiostoma ravennicum* **	**MFLUCC 14-0005**	**KP698415**	**KP698413**	**KP698414**	**N/A**	**N/A**
*Massarina corticola*	CBS 154.93	FJ795491	N/A	FJ795448	FJ795465	N/A
***Neoangustimassarina sichuanensis* ***	**CGMCC 3.20937**	**ON332917**	**ON332907**	**ON332925**	**ON383983**	**ON381175**
*Neoangustimassarina sichuanensis* *	UESTCC 22.0001	ON332918	ON332908	ON332926	ON383984	ON381176
** *Neooccultibambusa chiangraiensis* **	**MFLUCC 12-0559**	**KU712458**	**KU712442**	**KU764699**	**N/A**	**KU872761**
** *Neooccultibambusa kaiyangensis* **	**CGMCC 3.20404**	**MZ964886**	**MZ964868**	**MZ964877**	**OK061077**	**OK061071**
** *Neooccultibambusa trachycarpi* **	**CGMCC 3.20405**	**MZ964888**	**MZ964870**	**MZ964879**	**OK061079**	**OK061073**
** *Neothyrostroma encephalarti* **	**CBS 146037**	**N/A**	**MN562104**	**MN567612**	**N/A**	**MN556830**
** *Neothyrostroma encephalarti* **	**CPC 35999**	**N/A**	**MN562105**	**MN567613**	**N/A**	**MN556831**
** *Neovaginatispora clematidis* **	**MFLUCC 17–2156**	**MT226676**	**MT310606**	**MT214559**	**N/A**	**MT394738**
*Neovaginatispora fuckelii*	KH 161	AB618689	LC001731	AB619008	N/A	LC001749
*Neovaginatispora fuckelii*	KT 634	AB618690	LC001732	AB619009	N/A	LC001750
** *Occultibambusa aquatica* **	**MFLUCC 11-0006**	**KX698112**	**KX698114**	**KX698110**	**N/A**	**N/A**
** *Occultibambusa bambusae* **	**MFLUCC 13-0855**	**KU872116**	**KU940123**	**KU863112**	**KU940170**	**KU940193**
** *Occultibambusa chiangraiensis* **	**MFLUCC 16-0380**	**KX655551**	**N/A**	**KX655546**	**KX655566**	**KX655561**
** *Occultibambusa fusispora* **	**MFLUCC 11-0127**	**N/A**	**KU940125**	**KU863114**	**KU940172**	**KU940195**
** *Occultibambusa hongheensis* **	**KUMCC 21-0020**	**MZ329029**	**MZ329037**	**MZ329033**	**N/A**	**MZ325467**
** *Occultibambusa jonesii* **	**GZCC 16-0117**	**KY628324**	**N/A**	**KY628322**	**KY814758**	**KY814756**
** *Occultibambusa kunmingensis* **	**HKAS 102151**	**MT864342**	**MT627716**	**MN913733**	**MT878453**	**MT954407**
** *Occultibambusa maolanensis* **	**GZCC 16-0116**	**KY628325**	**N/A**	**KY628323**	**KY814759**	**KY814757**
** *Occultibambusa pustula* **	**MFLUCC 11-0502**	**KU872118**	**KU940126**	**KU863115**	**N/A**	**N/A**
***Occultibambusa sichuanensis* ***	**CGMCC 3.20938**	**N/A**	**ON332913**	**ON332931**	**ON383989**	**ON381181**
*Occultibambusa sichuanensis* *	UESTCC 22.0004	N/A	ON332914	ON332932	ON383990	ON381182
** *Palmiascoma gregariascomum* **	**MFLUCC 11-0175**	**KP753958**	**KP744452**	**KP744495**	**KP998466**	**N/A**
** *Palmiascoma qujingense* **	**KUMCC 19-0201**	**MT477186**	**MT477183**	**MT477185**	**MT495782**	**N/A**
** *Parapaucispora pseudoarmatispora* **	**KT 2237**	**LC100018**	**LC100021**	**LC100026**	**N/A**	**LC100030**
** *Paucispora quadrispora* **	**KT 843**	**AB618692**	**LC001734**	**AB619011**	**N/A**	**LC001755**
** *Paucispora versicolor* **	**KH 110**	**LC001721**	**AB918731**	**AB918732**	**N/A**	**LC001760**
** *Podocarpomyces knysnanus* **	**CBS 146076**	**N/A**	**MN562155**	**MN567662**	**MN556816**	**MN556836**
** *Polyplosphaeria fusca* **	**KT1616**	**AB524463**	**AB524789**	**AB524604**	**N/A**	**N/A**
** *Pseudotetraploa curviappendiculata* **	**JCM 12852**	**AB524467**	**AB524792**	**AB524608**	**N/A**	**N/A**
*Pseudotetraploa javanica*	JCM 12854	AB524470	AB524795	AB524611	**N/A**	**N/A**
** *Pseudotetraploa longissima* **	**JCM 12853**	**AB524471**	**AB524796**	**AB524612**	**N/A**	**N/A**
** *Pseudotetraploa rajmachiensis* **	**NFCCI 4618**	**N/A**	**MN937222**	**MN937204**	**N/A**	**N/A**
***Pseudotetraploa bambusicola* ***	**CGMCC 3.20939**	**ON332923**	**ON332915**	**ON332933**	**ON383991**	**ON381183**
*Pseudotetraploa bambusicola* *	UESTCC 22.0005	ON332924	ON332916	ON332934	ON383992	ON381184
** *Quadricrura bicornis* **	**CBS 125427**	**AB524472**	**AB524797**	**AB524613**	**N/A**	**N/A**
** *Quadricrura meridionalis* **	**CBS 125684**	**AB524473**	**AB524798**	**AB524614**	**N/A**	**N/A**
** *Seriascoma bambusae* **	**KUMCC 21-0021**	**MZ329031**	**MZ329039**	**MZ329035**	**MZ325470**	**MZ325468**
** *Seriascoma didymosporum* **	**MFLUCC 11-0179**	**KU872119**	**KU940127**	**KU863116**	**KU940173**	**KU940196**
** *Seriascoma yunnanense* **	**MFLU 19-0690**	**MN174694**	**N/A**	**MN174695**	**MN210324**	**MN381858**
** *Shrungabeeja aquatica* **	**MFLUCC 18-0664**	**N/A**	**MT627722**	**MT627663**	**N/A**	**N/A**
** *Shrungabeeja longiappendiculata* **	**BCC 76463**	**KT376471**	**KT376474**	**KT376472**	**N/A**	**N/A**
*Shrungabeeja vadirajensis*	MFLUCC 17-2362	N/A	MT627681	MN913685	**N/A**	**N/A**
** *Tetraploa aquatica* **	**MFLU 19-0995**	**N/A**	**MT530448**	**MT530452**	**N/A**	**N/A**
*Tetraploa aristata*	CBS 996.70	AB524486	AB524805	AB524627	**N/A**	**N/A**
** *Tetraploa dwibahubeeja* **	**NFCCI 4621**	**N/A**	**MN937226**	**MN937208**	**N/A**	**N/A**
** *Tetraploa nagasakiensis* **	**JCM 13168**	**AB524489**	**AB524806**	**AB524630**	**N/A**	**N/A**
** *Triplosphaeria acuta* **	**JCM 13171**	**AB524492**	**AB524809**	**AB524633**	**N/A**	**N/A**
** *Triplosphaeria maxima* **	**JCM 13172**	**AB524496**	**AB524812**	**AB524637**	**N/A**	**N/A**
** *Triplosphaeria yezoensis* **	**CBS 125436**	**AB524497**	**AB524813**	**AB524638**	**N/A**	**N/A**
** *Vaginatispora amygdali* **	**CBS 143662**	**LC312495**	**LC312524**	**LC312553**	**LC312611**	**LC312582**
** *Vaginatispora appendiculata* **	**MFLUCC 16-0314**	**KU743219**	**KU743217**	**KU743218**	**N/A**	**KU743220**
*Vaginatispora aquatica*	MFLUCC 11-0083	KJ591575	KJ591577	KJ591576	N/A	N/A
** *Versicolorisporium triseptatum* **	**JCM 14775**	**AB524501**	**AB365596**	**AB330081**	**N/A**	**N/A**
*Versicolorisporium triseptatum*	UESTCC 21.0016 = NMX1222	OL741381	OL741378	OL741318	N/A	N/A

**Table 3 jof-08-00720-t003:** Morphological comparative data of *Flabellascoma* species.

Taxa	Ascomata (μm)	Hamathecium (μm)	Asci (μm)	Ascospores (μm)	Sheath (μm)	References
*F. aquaticum*	280–440 × 260–390	1.2–2.0	48.0–72.0 × 8–9	16–18 × 4.3–5.3	4.7–7.0 μm wide	[57]
*F. cycadicola*	490–530 × 600–620	1.0–3.0	67.5–88.0 × 9–12	17–23 × 4.5–7.0	7.0–10 μm long	[56]
*F. fusiforme*	310–420 × 320–380	1.5–3.0	66.0–80.0 × 10–12	15–18 × 4.0–5.0	5.4–8.0 μm wide	[57]
*F. minimum* ^T^	250–320 × 350–500	1.5–3.0	45.0–77.5 × 7.5–12	12–17.5 × 3.5–5	5.5–8.0 μm long	[56]
** *F. sichuanense* **	**150–350 × 190–280**	**1.5–3.5**	**55.0–75.0 × 9.5–13**	**15–18 × 5.0–7.0**	**3.0–7.0 μm long, 1.5–2.5 μm wide**	**This study**

**Table 4 jof-08-00720-t004:** Morphological comparative data of *Occultibambusa*.

Taxa	Ascomata (μm)	Asci		Ascospores		References
		Morphology	Size (μm)	Morphology	Size (μm)	
** *O. aquatica* **	**100–250 × 180–280**	Clavate	73–86 × 9–13	Narrowly fusiform, 1-septate, brownish, with a sheath	19–25 × 3.5–6.5	[62]
*O. bambusae* ^T^	150–200 × 400–550	Broadly cylindrical	(50–)60–80(−90) × (9.5–)11.5–14.5(−15)	Slightly broad and fusiform, 1-septate, dark brown, with a sheath	(22–)23.5–27.5× 4.5–7	[47]
*O. chiangraiensis*	195–295 × 352–520	Clavate-oblong	47–92 × 12–16	Fusiform, (1–)3-septate, hyaline when immature, pale brown to red-brown at maturity, without a sheath	16–24 × 5–7	[62]
*O. fusispora*	135–185 × 240–275	Clavate to cylindric-clavate	(60–)65–90(−110) × (11–)12–14(−15)(−16)	Fusiform, mostly 1-septate, rarely 2–3-septate, light brown, without a sheath	(20–)22–25(−26) × 5–6(−6.5)	[47]
*O. hongheensis*	180–340 × 400–550	Cylindric-clavate to clavate	(78–)80–130(–137) × (18–)19–23(–25)	Fusiform, 1-septate, hyaline when young and becoming pale brown when mature, with a sheath	(25–)27–30 × (5.5–)8–9(–10)	[60]
*O. jonesii*	196–236 × 200–260	Broadly cylindrical to clavate	(65–)75–89(–105) × 13.5–19	Inequilateral-fusiform, 2-celled, hyaline when young and becoming brown to grayish when mature, without a sheath	27–33.5 × 5.5–6.5	[63]
*O. kunmingensis*	110–150 × 220–260	Clavate or cylindric-clavate	110–140(–160) × 13–16.5	Fusiform, 1-septate, brown, without a sheath	32–40 × 5–6.5	[52]
*O. maolanensis*	544–600 diameter	Broadly cylindrical to clavate	(66–)77–85(–94) × 17–20(–24)	Inequality-fusiform, 2-celled, hyaline when young and become light brown when mature, without a sheath	25–31 × 8–10	[63]
*O. pustula*	150–200 × 200–300	Cylindrical	80–105 × 8–12	Slightly broad-fusiform, 1-septate, hyaline to pale brown, with a sheath	22–25 × 5–5.5	[47]
** *O. sichuanensis* **	**130–180 ×** **340–440**	**Obovoid to pyriform**	**70–100 × 22–27**	**Fusiform, 1-septate, brown, with a sheath**	**31–41 × 6.5–10**	**This study**

## Data Availability

The sequences data were submitted to GenBank.

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
