# Peer review of "Morpho-Phylogenetic Evidence Reveals Novel Pleosporalean Taxa from Sichuan Province, China"

_jof, 2022, doi:10.3390/jof8070720_

Round 1

Reviewer 1 Report

Several items must be addressed in the paper:

1.  A native English speaker must review this paper and edit the grammar.

2.  ALL new taxa are INVALIDLY published.  Please read the ICTF paper on How to publish a new fungal species, or name, version 3.0 in IMAFungus and redo the part where you designate the HOLOTYPE.

3.  Include a section on culture methods.

4.  Number figures in alphabetical order.

Author Response

Dear Editor,

Thanks very much for your efforts and reviewers’ comments on our manuscript. We have revised our paper following the comments and suggestions.

  • The formal new taxa enablements have been checked and revised following the guideline paper suggested by the reviewer.
  • we checked the References in-text and end-list for uniformity in style.
  • we also did minor revisions and highlighted them in the revised file.

Herein we provide our response to the reviewers’ comments point by point. Please kindly see the details below. Thank you very much.

Sincerely yours

Jian-Kui Liu

Response to reviewer’s comments

Note: The line number listed here are based on ‘the review version’.

Reviewer #1: A native English speaker must review this paper and edit the grammar.

Our response: Thanks for the suggestion. We have checked and improved the parts where necessary.

Reviewer #1: ALL new taxa are INVALIDLY published.  Please read the ICTF paper on How to publish a new fungal species, or name, version 3.0 in IMA Fungus and redo the part where you designate the HOLOTYPE.

Our response: Thanks for your comments. We have corrected the parts where we designated the HOLOTYPE and ISOTYPE. Please kindly see the revised version.

Reviewer #1: Include a section on culture methods.

Our response: We added isolation and culture methods. Please kindly see the revised version.

Reviewer #1: Number figures in alphabetical order.

Our response: We have corrected it, please see the Figure 3 and its legend in the revised version.

Reviewer #1: Line 26-28: sentence too long!

Our response: We re-wrote the sentence and highlighted it, please kindly see the revised version Line 26-29.

Reviewer #1: Line 49-50: Not a sentence!

Our response: We re-wrote this sentence. Please kindly see the revised version Line 50-52.

Reviewer #1: Line 56-61: delete, what are the goals of this study?

Our response: We deleted this sentence. The goal of this study is to describe the new findings and contribute fungal diversity to China.

Reviewer #1: Line 79: How did you isolate fungi in culture? There are no methods for fungal culture isolation.

Our response: Thank you for pointing out this. It is the same comment mentioned above. We added it in the “Material and Methods” section.

Reviewer #1: Line 123: Why an odd number of generations?  How did you know to stop the run? When did it reach stationarity?

Our response: We read the relevant information from the log file (see the screenshot below). The average standard deviation of split frequencies (<0.01) was set in commend to stop the run, and the analysis stopped when the convergence diagnostic hit the stop value.

Reviewer #1: Figure 1: Why is this yellow color here? Why is this branch here?

Our response: The yellow color is because of our carelessness during the tree editing. The branch represents the “root length” of the tree. We provided a newly edited tree instead. Please see the revised version.

Reviewer #1: Line 147: use present tense, not past.

Our response: Thanks for the kind advice. We corrected it in the present tense.

Reviewer #1: Line 172: This should be moved up and just below the new genus name.

Our response: We have corrected it according to your suggestion.

Reviewer #1: Line 180: Some may not accept this as a new genus. It could be considered another species of Angustimassarina and Podocarpomyces could be a synonym of Angustimassarina.

Our response: Thank you very much. We had the same concern when we first got the result. However, we decided to introduce it as a new genus after we compare the morphology and phylogeny. The main evidence is that the known asexual morph of Angustimassarina populi (type species of Angustimassarina) and Podocarpomyces are distinct (hyphomycetous with elongate clavate, pale to dark brown, 1–3-septate conidia vs. coelomycetous with aseptate, hyaline conidia). On the other hand, the sexual morph of P. knysnanus (type species of Podocarpomyces) is undetermined, which also makes it difficult to link Angustimassarina and Podocarpomyces. The new genus Neoangustimasssarina we introduced in this study is justified by morphology and multi-gene phylogeny. In addition, we also compare the single gene of these three genera and they can be recognized as different genera.

Reviewer #1: Line 180, 217: This name will be INVALID as published! Read the ICTF paper on how to publish a new species and follow the guidelines.

Our response: These are the same comments mentioned above. Please kindly see our response above.

Reviewer #1: Line 235: What do the arrows point to? You need to letter the photos in alphabetical order!

Our response: The arrows indicate how the conidiogenous cells produce conidia. We lettered the photos in alphabetical order in Figure 3, please kindly see the revised version.

Reviewer #1: Line 296, 318: Why not call this an ascomal wall?

Our response: The “ascomal wall” and “peridium” are the same meaning. In the dictionary of the fungi, “peridium” means the wall or limiting membrane of a sporangium or other fruit-body. We prefer to use “peridium” herein.

Reviewer #1: Line 352: Versicolorisporium is not a monophyletic genus - it is nested in Occultibambusa!

Our response: We discussed this in the discussion section. Please kindly see the revised version.

Reviewer #1: Line 398-405: delete, why are you even discussing this genus?  It has nothing to do with your study!

Our response: Thanks for your comments. We deleted this paragraph, please kindly see the revised version.

Reviewer #1: Line 424: What type species issue?  Explain.

Our response: We would like to say that the type species of Occultibambusa and Versicolorisporium only have sexual morph and asexual morph, respectively. The synonym issue (genus)can only be applied based on the type species, and we have explained this in the following sentence “Therefore, further studies are needed to provide sexual and asexual links of the type species of O. bambusae and V. triseptatum towards the classification of Occultibambusa and Versicolorisporium with more sampling and taxa population included in the analysis” in the paragraph.

Reviewer 2 Report

I recommend expanding: Introduction, discussion, and conclusions sections with more updated literature.

The authors are advised to improve the manuscript in terms of adequate language levels as well as research paper structure.

The authors should elaborate more on their findings and discussion compared to other studies, to their importance.

Please check the References in-text and end-list for uniformity in style.

The conclusion you have provided is quite brief and provides sufficient feedback on the main objectives of your study.

Author Response

Dear Editor,

Thanks very much for your efforts and reviewers’ comments on our manuscript. We have revised our paper following the comments and suggestions.

  • The formal new taxa enablements have been checked and revised following the guideline paper suggested by the reviewer.
  • we checked the References in-text and end-list for uniformity in style.
  • we also did minor revisions and highlighted them in the revised file.

Herein we provide our response to the reviewers’ comments point by point. Please kindly see the details below. Thank you very much.

Sincerely yours

Jian-Kui Liu

Response to reviewer’s comments

Note: The line number listed here are based on ‘the review version’

Reviewer #2: I recommend expanding: The introduction, discussion, and conclusions sections with more updated literature.

Our response: We considered your kind suggestions and improved some sentences in those parts where necessary. Please kindly read the revised version.

Reviewer #2: The authors are advised to improve the manuscript in terms of adequate language levels as well as research paper structure.

Our response: We have improved the grammar of the manuscript. Please kindly read the revised version.

Reviewer #2: The authors should elaborate more on their findings and discussion compared to other studies, to their importance.

Our response: Thanks very much for your comments. We have discussed Sichuan’s unique natural conditions, which have great potential for the excavation and identification of fungi. As a fungal taxonomy and phylogeny paper, the main aim of this study is to describe the new taxa and contribute the fungal diversity to China.

Reviewer #2: Please check the References in-text and end-list for uniformity in style.

Our response: Thank you very much for your kind advice. We have carefully checked it according to your suggestion.

Reviewer #2: The conclusion you have provided is quite brief and provides sufficient feedback on the main objectives of your study.

Our response: Thanks very much for your comments. As we mentioned above, this paper is a typical fungal taxonomy and phylogeny study, the main aim is to describe the new taxa with morphology and phylogeny evidence, and contribute the fungal diversity to China. This has been given in the “Discussion” part as a brief conclusion.

Round 2

Reviewer 1 Report

say branches are thickened, not bold!

"...probabilities (PP) from MCMC analysis equal to or greater than 0.95 were in bold. "